

# Transcriptomic analyses provide new insights into green and purple color pigmentation in *Rheum tanguticum* medicinal plants

Haixia Chen[1,2,*], Tsan-Yu Chiu[2,*], Sunil Kumar Sahu[2,3], Haixi Sun[2], Jiawen Wen[2], Jianbo Sun[4], Qiyuan Li[4], Yangfan Tang[5], Hong Jin[6] and Huan Liu[1,2,3]

[1] College of Life Sciences, University of Chinese Academy of Sciences, Beijing, China
[2] BGI-Shenzhen, Beishan Industrial Zone, Shenzhen, China
[3] State Key Laboratory of Agricultural Genomics, BGI-Shenzhen, Shenzhen, China
[4] China National GeneBank, BGI-Shenzhen, Jinsha Road, Shenzhen, China
[5] Sichuan Academy of Chinese Medicine Sciences, Chengdu, Sichuan, PR China
[6] Fairy Lake Botanical Garden, Shenzhen & Chinese Academy of Sciences, Shenzhen, China
[*] These authors contributed equally to this work.

Corresponding authors
Hong Jin, jinhongbio@qq.com
Huan Liu, liuhuan@genomics.cn

## ABSTRACT

**Background**. *Rheum tanguticum Maxim. ex Balf* is a traditional Chinese medicinal plant that is commonly used to treat many ailments. It belongs to the Polygonacae family and grows in northwest and southwest China. At high elevations, the color of the plant's young leaves is purple, which gradually changes to green during the growth cycle. Anthraquinone, which is known for various biological activities, is the main bioactive compound in *R. tanguticum*. Although a significant amount of research has been done on *R. tanguticum* in the past, the lack of transcriptome data limits our knowledge of the gene regulatory networks involved in pigmentation and in the metabolism of bioactive compounds in *Rheum* species.

**Methods**. To fill this knowledge gap, we generated high-quality RNA-seq data and performed multi-tissue transcriptomic analyses of *R. tanguticum*.

**Results**. We found that three chlorophyll degradation enzymes (*RtPPH*, *RtPao* and *RtRCCR*) were highly expressed in purple samples, which suggests that the purple pigmentation is mainly due to the effects of chlorophyll degradation. Overall, these data may aid in drafting the transcriptional network in the regulation and biosynthesis of medicinally active compounds in the future.

## INTRODUCTION

*Rheum tanguticum Maxim. ex Balf* is a traditional Chinese medicinal plant belonging to the Polygonaceae family. Because its young leaves are shaped like chicken feet, *R. tanguticum* is also known as "Tangut *Rheum*" or "chicken feet *Rheum*" (*Wang, 2009*). The roots and rhizomes of this species are commonly called Chinese rhubarb, and are generally used as traditional Chinese medicine due to their strong antibacterial (*Lu et*

*al., 2011*), antineoplastic (*Huang et al., 2007*), and anti-inflammatory (*Choi et al., 2013*) effects. National Medical Products announced Lian-hua-qing-wen (LHQW) as a certified traditional Chinese formulation to treat fever, cough, or fatigue caused by the mild or common types of COVID-19 in the "Pharmaceutical Supplement Application Document" (*Chen et al., 2021*). As one of the LHQW ingredients, rhein from Dahuang Rhei Radix et Rhizoma extracts was identified as having potential ACE2 binding activity (*Chen et al., 2021*). Because of its medicinal importance, *Rheum* is facing overexploitation (*Zhou et al., 2014*; *Wang et al., 2016*), leading to such a rapid decline in wild *Rheum* that it is now considered endangered (*Yang et al., 2001*). To facilitate conservation, previous studies have provided preliminary assessments on the genetic variation of wild *R. tanguticum* using SSR and ISSR analyses (*Chen et al., 2009*; *Hu et al., 2010*; *Wang et al., 2012*). In addition, a karyotype analysis showed that *R. tanguticum* is a diploid (2n = 22) and no polyploidy was found (*Yanping, Wang & Li, 2011*). The genetic diversity of *R. tanguticum* has also been reported, but based on very limited samples (only collected from the Qinghai-Tibet Plateau (*Chen et al., 2009*) or the Qinghai province *Hu et al., 2010*).

Plant pigments are vital in signaling, protecting, and determining the colors of plants (*Lee, 2005*). These pigments can be classified into four categories: chlorophylls, anthocyanins, carotenoids, and betacyanins (*Dikshit & Tallapragada, 2018*). Anthocyanins, carotenoids, and betacyanins are responsible for the natural red colors found in plants (*Fernández-López, Fernández-Lledó & Angosto, 2020*). Anthocyanin can generate many different colors, such as red, purple, blue, yellow, and orange (*Castañeda Ovando et al., 2009*). Chlorophyll plays an important photosynthetic role in plants, contributing to plant growth (*Li et al., 2018*) as well as the characteristic green color (*Croft & Chen, 2018*) of plants. A previous study on *Camellia sinensis* (L.) O. Kuntze, a cultivated tea with purple young leaves and green mature leaves, showed that the purple tea leaves have higher levels of anthocyanin, total polyphenols, and total catechins, but lower chlorophyll, carotenoid, and soluble sugar (*Zhou, Sun & Lai, 2016*).

*R. tanguticum* is known to grow under harsh environmental conditions such as low atmospheric pressure, low temperature, and high solar radiation. Plants produce vast and versatile phytochemical constituents which play key roles in mediating plant–environment interactions. According to field observations, when *R. tanguticum* grows at high elevations, it generates purple leaves that progressively turn green as the plant matures. However, no color changes are observed when these plants are cultured at lower elevations. Another Rheum species, *Rheum palmatum* L, which has a close phylogenetic relationship with *R. tanguticum*, has large, rough, palmate leaves that are greenish-purple in color (https://www.botanical-online.com/en/botany/rhubard-chinese). Recently, a comparative transcriptome analysis of *Rheum austral* was done to understand the adaptive strategies of *R. australe* in its niche (*Mala et al., 2021*). Despite previous studies, a scarcity of transcriptome data hinders our understanding of the gene regulatory networks involved in the different pigmentations at high elevation and the metabolism of bioactive compounds in *Rheum* plants. Here, we present another valuable RNA-seq data set to establish an initial regulatory network in *R. tanguticum* and to help fill these knowledge gaps.

## MATERIALS & METHODS

### Plant materials, sample collection, and preparation of total RNA

Fresh tissues from both purple and green *R. tanguticum* (Voucher Number 51322420120805441) were collected from the Aba Tibetan and Qiang autonomous prefecture (33°68′N, 103°25′E) located in the Sichuan province of China by Yangfan Tang (Sichuan Academy of Chinese Medicine Sciences) and taxonomically identified by Qingmao Fang (Sichuan Academy of Chinese Medicine Sciences). The collection of plant material in this study complied with all relevant institutional, national, and international guidelines and legislation. The samples were flash-frozen in liquid nitrogen on-site. A total of 19 *R. tanguticum* samples were collected from different parts of *R. tanguticum* plants: five green leaf biological duplications, five green petiole biological duplications, two green rhizome biological duplications, four purple leaf biological duplications, and three purple petiole biological duplications. Total RNA was extracted with the CTAB-pBIOZOL (CAT# BSC55M1), according to the manufacturer's instructions.

The statistical power of this experimental design, calculated in RnaSeqSampleSize (https://github.com/slzhao/RnaSeqSampleSize), was 0.7.

### Quality evaluation of total RNAs, library preparation, and sequencing

The evaluations of the RNA samples were carried out using Qubit 2.0, Nanodrop, and Agilent 2100 (*Simbolo et al., 2013*) to ensure that the concentration, integrity, and purity were suitable for library preparations and RNA sequencing. The samples with an RNA integrity number (RIN) value over seven were moved to library preparation using the MGIEasy RNA kit (CAT# 1000006383). The constructed library was used for sequencing by the BGISEQ-500 (*Huang et al., 2018*) platform.

### Pre-processing of raw reads and *de novo* transcriptome assembly

We used FastQC (version 0.11.3) (*Andrews, 2010*) to confirm the validation of the raw data and the low quality reads were filtered out using Trimmomatic (version 3) (*Bolger, Lohse & Usadel, 2014*). We then performed a second quality validation of just the clean reads, using FastQC, to ensure they were suitable for downstream analyses. All FastQC results were visualized using MultiQC (version: 1.9) (*Ewels et al., 2016*). Clean reads were mapped to the unigene using Bowtie2 (version 2.2.5) (*Langmead & Salzberg, 2012*). The Trinity (*Haas et al., 2013*) (v2.9.1) software was used to assemble the short k-mers into contigs based on the de Bruijin Graph algorithm. The sequencing depth of all used data was 678 ×.

### Transcriptome annotation

We used the TransDecoder (v3.0.1; http://transdecoder.sourceforge.net) to predict the potential coding sequence of the unigenes. First, the longest open reading frame (ORF) was selected from the transcript sequences and then scanned with known protein sequences (Swiss-Prot 2020 database and Pfam-A.hmm 2020 database) using blastp and hmmscan. These results were then combined to predict the coding region of *R. tanguticum*.

The assembled unigene sequences were functionally annotated by aligning them with the Kyoto Encyclopedia of Genes and Genomes (KEGG) (*Kanehisa & Goto, 2000*), Gene

Ontology (GO) (*Ashburner et al., 2000*; *The Gene Ontology, 2021*), clusters of orthologous groups for eukaryotic complete genomes (KOG) (*Tatusov et al., 2000*), SwissProt (*Bairoch & Apweiler, 2000*), Pfam (http://pfam.xfam.org/), and National Center for Biotechnology Information (NCBI) non- redundant (NR) protein databases, as well as the Nucleotide Sequence Database (NT) with BLAST ( *E*-value $\leq$ 1e–05).

## Differential gene expression analysis

The RSEM pipeline was used to determine the FPKM (fragments per kilobase per million) value of different samples. The clean data were re-mapped to the assembled transcriptome using bowtie2 (version 2.2.5) (*Langmead & Salzberg, 2012*). Bowtie-build was used to make a reference library. The ''–transcript-to-gene-map'' parameter was then used to map the transcripts to the corresponding genes. Filtered sequencing reads were mapped to the reference by bowtie2 and then the ''rsem-calculate-expression'' option was used to quantify the expression level. A differential expression analysis was carried out with the DESeq2 (version 1.28.1) (*Love, Huber & Anders, 2014*) package in R (version 4.0.2). The screening criteria for differential expression genes were: adjusted pvalue <0.05, log2FoldChange >1 (up-regulated) or log2FoldChange <−1 (down-regulated). The information of the grouping and control samples from the differential expression gene analysis is summarized in Table S1. Based on the DEseq2 normalized data matrix, we calculated the Pearson correlation coefficients between different samples. The co-expression analysis of anthraquinone pathway genes and TFs was performed using the WGCNA R package.

## Ethics approval

This study, including sample collection, was conducted according to the ethical clearance of the 10,000 Plant Genomes Project—10KP (NO. FT20026) by the institutional review board of BGI which permits the use of biological resources for scientific research purposes.

## RESULTS

### RNA data quality assessment and *de novo* assembly

In *R. tanguticum*, we constructed short-read RNA-seq libraries and a total of 221.09 Gb raw data and 213.94 Gb high-quality data were generated for quantification, and differential gene expression analyses. The percentage of clean reads ranged from 95% to 99%, the mapping rates were about 70% in *R. tanguticum*, and the Q20 value of the clean reads was around 90%. The clean data from a total of 19 samples were subjected to further transcriptome assembly. We retrieved 336,987 transcripts and 120,261 unigenes with contig N50 1,761 bp and contig N50 1,474 bp, respectively. The completeness of the transcriptome was evaluated using the Benchmarking Universal Single-Copy Ortholog (BUSCOs) resulting in 91.9% (S:25.5%, D:66.4%, F:5.7%, M:2.4%) coverage. The reads number, clean reads percentage, mapping rate, Q20 value, and GC content of the data were assessed and the results are summarized in Table S2.

The PCA analysis showed that the samples in the dataset clustered into four major groups corresponding to the tissue types and plant colors (Fig. 1A). PC1 and PC2 explained 44%

and 33% of the total variance in gene expression of the *R. tanguticum* data set, respectively. The heatmap clustering of Pearson correlation coefficients from the comparison of all 19 samples with various tissue types revealed a strong correlation (0.81–1) between replicates (Fig. 1B). Taken together, these results suggest that our datasets are a reliable data resource for future studies.

## Functional annotation

A total of 45,866 out of the 120,261 total unigenes (38.14%) showed significant similarities to known proteins in *R. tanguticum* (Table S3). Gene Ontology (GO) assignments were used to classify the unigene sequences based on functional annotation to determine their possible functions in *R. tanguticum*. There were 27,423 unigenes that could be categorized into functional groups under the "cellular component," "molecular function," and "biological process," divisions in *R. tanguticum* (Fig. 2A). For the biological process group, genes involved in the "cellular process" (12,518) and "metabolic process" (10,908) were the most highly represented. For unigenes in the cellular component group, "cellular anatomical entity" (14,987) and "intracellular" (8,183) were the most highly represented categories, followed by "protein-containing complex" (2,519). For the molecular function group, "binding" (15,434), "catalytic activity" (12,960), and "transporter activity" (1,535) were the most represented categories.

A functional characterization analysis based on the KEGG database was performed on the unigenes generated in the present study. In summary, the most represented pathways in *R. tanguticum* were: "global and overview maps" (9,223), "carbohydrate metabolism" (4,170), "translation" (2,960), and "folding, sorting and degradation" (2,172) (Fig. 2B). The identified transcripts enriched in these diverse metabolic pathways will help us better understand the active ingredients in *R. tanguticum*.

## Transcriptome expression and differentially expressed genes of *R. tanguticum*

The FPKM values of all unigenes in different samples were summarized in Table S4. To investigate the different colors between the leaves and petioles of *R. tanguticum*, we calculated the expression levels of the unigenes between these two tissues. Comparing green leaves (GL) and purple leaves (PL) showed 4,861 DEGs in total, with 2,426 up- and 2,435 down-regulated unigenes, respectively. Further, we did a KEGG enrichment analysis for these DEGs. Among the down-regulated genes, 51 genes were assigned to "photosynthesis," 48 genes clustered in "carbon fixation in photosynthetic organisms," and 22 genes were assigned to "photosynthesis –antenna proteins" (Fig. 3A). Interestingly, the unigenes enriched in secondary metabolism, such as the "flavone and flavonol biosynthesis" (eight) and "sesquiterpenoid biosynthesis" (19) pathways, were up-regulated (Figs. 3A; 3C). We also compared the green petioles and purple petioles and found 3,028 up- and 2,413 down-regulated unigenes. Notably, there were no up- or down-regulated unigenes associated with photosynthesis-related pathways. Three major up-regulated genes were enriched in "metabolic pathways" (694), "biosynthesis of secondary metabolism" (346), and "amino sugar and nucleotide sugar metabolism" (120). These results suggest that the expression profiles of the petiole are different than the expression profiles of the leaves (Figs. 3B; 3D).

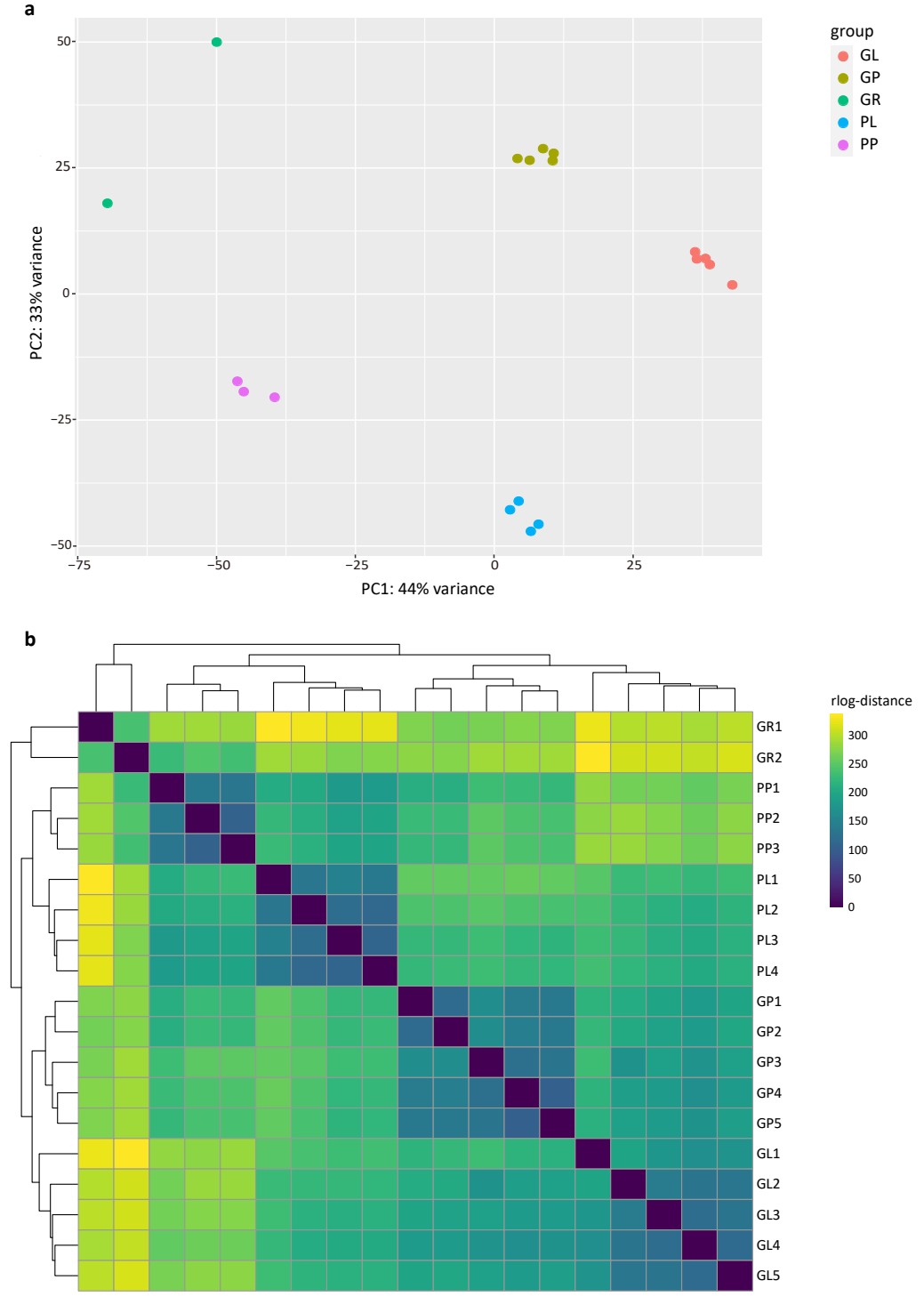

**Figure 1** **Exploratory analysis of the correlation between RNA-seq samples.** (A) PCA plot of 19 *R. tanguticum* samples. (B) Heatmap clustering of correlation coefficients across 19 samples in *R. tanguticum*. GL, green leaf; GR, green rhizome; GP, green petiole; PL, purple leaf; PP, purple petiole.

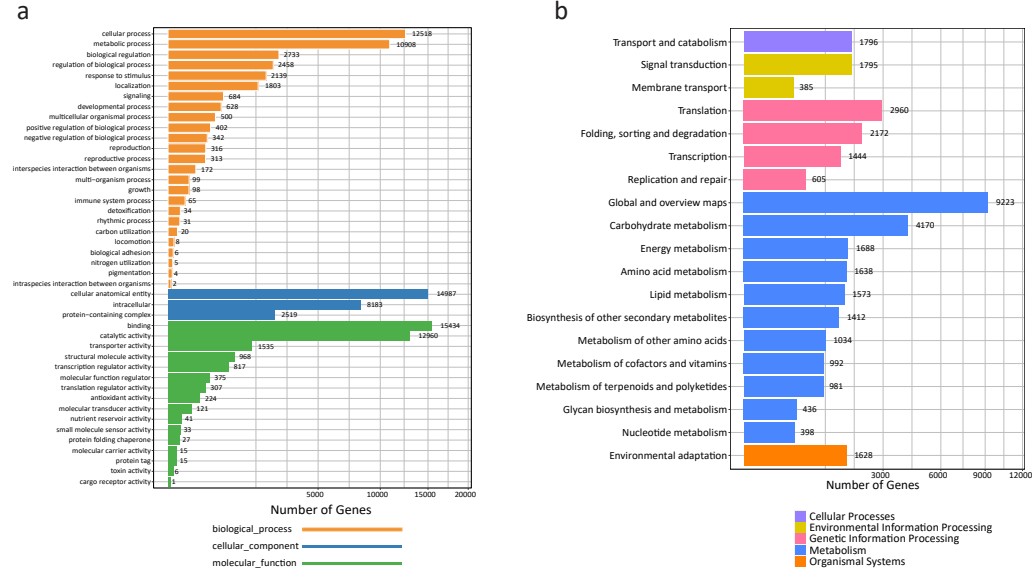

**Figure 2** **GO and KEGG annotation of *R. tanguticum* unigenes.** (A) GO annotation of all the *R. tanguticum* unigenes. (B) KEGG annotation of all the *R. tanguticum* unigenes. Representative and important pathways are presented in this figure.

We also performed a GO enrichment analysis. In this analysis, many down-regulated unigenes between the green and purple leaves were also found to be enriched in photosynthesis-related pathways, such as "photosystem II stabilization" (five), "photosystem II assembly" (six), "photosystem I reaction center" (10), and "photosynthesis, light harvesting" (23) (Fig. 4; Table S5). The GO enrichment analysis of up-regulated unigenes between GL and PL found that they were enriched in cell cycle-associated genes including "microtubule-based process" and "nucleic acids metabolisms process" (Table S6). The up-regulated genes in the petioles (GP vs. PP) were enriched in cell wall metabolism and development-associated genes (Table S7), while the down-regulated genes in the petioles were enriched in amino acids transport and cell wall catabolism (Table S8).

To compare tissue-specific expression patterns, we compared the expression profiles of the leaves and petioles of the same colors. The comparative transcriptomic study between GL and GP found 2,004 up-regulated and 1,690 down-regulated DEGs. The enrichment analysis results of these DEGs are summarized in Tables S9 and S10. By comparing the PL and PP, we found 2,687 up-regulated and 1,745 down-regulated DEGs. The results of the KEGG and GO enrichment analyses of the down- and up-regulated DEGs between PL and PP are summarized in Tables S11 and S12.

## Comparing the expression of unigenes involved in chlorophyll and anthocyanin in *R. tanguticum*

Chlorophyll is the most abundant pigment on earth and is a key component of photosynthesis required for the absorption of sunlight (*Hörtensteiner & Kräutler, 2011*). We

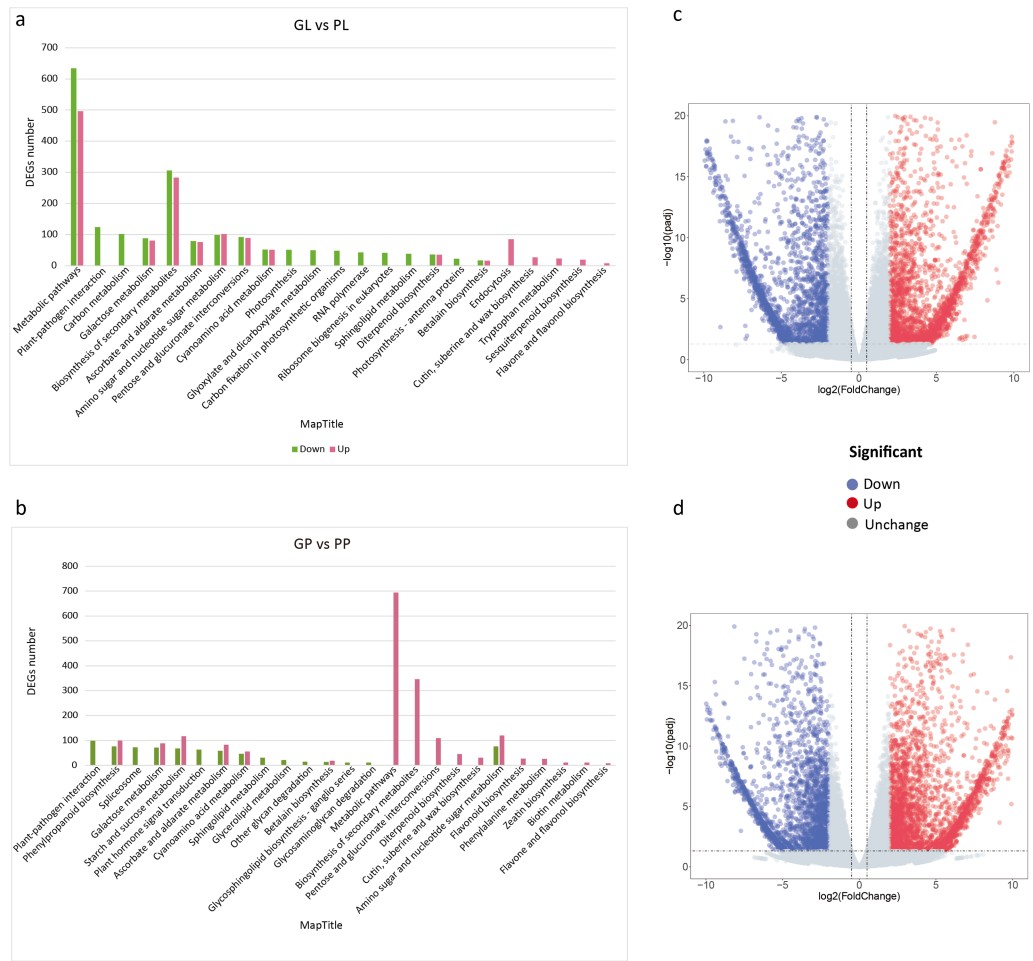

**Figure 3** **KEGG pathway enrichment of DEGs between purple and green of _R. tanguticum_.** (A) KEGG pathway enrichment of DEGs between green and purple leaf. (B) KEGG pathway enrichment of DEGs between the green petiole and purple petiole. Green color represents down DEGs, pink color represents up DEGs. (C) Volcano plots of the transcriptome between green and purple leaf. (D) Volcano plots of the transcriptome between green and purple petiole.

identified chlorophyll pathway genes (Table S13) and then constructed the expression maps of those genes. Interestingly, some enzyme genes (_RtGSA, RtHEM, RtHEMC, RtHEME, RtHEMF, RtHEMG, RtCAO, RtNYC1/RtNOL, RtCLH, RtSGR_) were highly expressed in green samples. In contrast, we found that genes involved in chlorophyll degradation like _RtPPH_ (pheophytinase), _RtPaO_ (pheophorbide a oxygenase) and _RtRCCR_ (red chlorophyll catabolite reductase) exhibited higher expression levels in purple samples (Fig. 5). The purple color of _R. tanguticum_ could be due to the combination of lower chlorophyll biosynthesis expression and higher _RtPPH, RtPaO,_ and _RtRCCR_ expression in the chlorophyll degradation pathway.

Anthocyanin is a major group of plant pigments that may appear red, purple, blue, or black in various tissues. The analysis of the _R. tanguticum_ transcriptome data set revealed
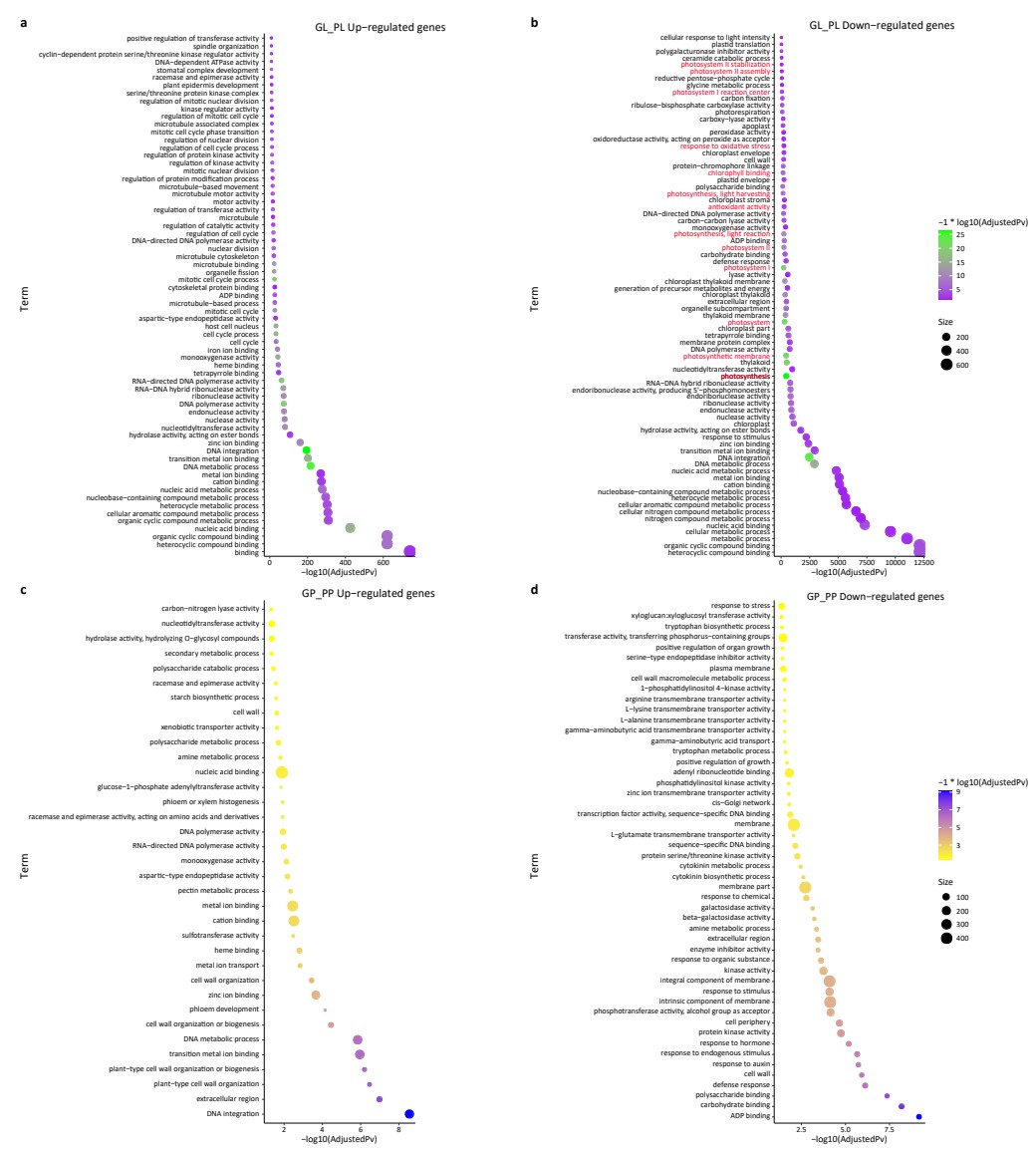

**Figure 4** **GO enrichment of DEGs between green and purple samples of *R. tanguticum*.** (A) The GO enrichment terms about up DEGs between green leaf (GL) and purple leaf (PL). (B) The GO enrichment terms about down DEGs between green leaf (GL) and purple leaf (PL). Genes associated with photosynthesis is labeled in red. (C) The GO enrichment terms about up DEGs between the green petiole (GP) and purple petiole (PP). (D) The GO enrichment terms about down DEGs between the green petiole (GP) and purple petiole (PP).

that 73 unigenes exerted a direct influence over 10 enzymes that are known to be involved in the anthocyanin pathway, and almost all of those 73 unigenes were demonstrated to be a multigene family. Anthocyanidin synthase (ANS) and Anthocyanidin 3-O-glucoside 2'''-O-xylosyltransferase (UFGT) catalyze the last two steps of anthocyanidin biosynthesis and are therefore key enzymes in the biosynthesis of blue or red pigments (*Chen et al., 2011*;

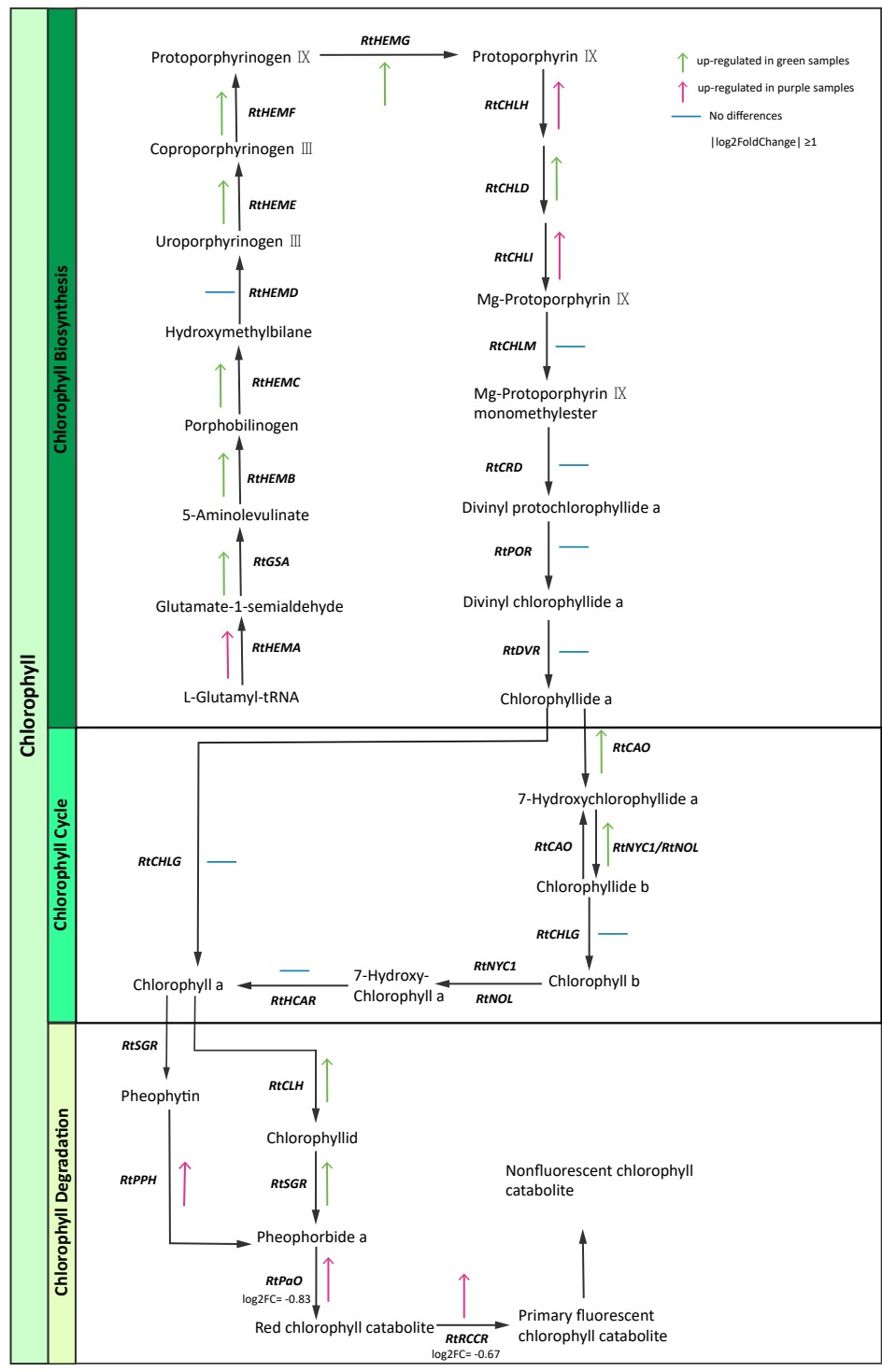

**Figure 5** **Chlorophyll candidate pathway in *R. tanguticum*.** Green and purple arrow represent up-regulated in green and purple tissues of *R. tanguticum*, respectively. The transverse line means no differences between green and purple samples.

*Zhang et al., 2020*). However, these two genes were more highly expressed in green samples compared to the purple tissues, which did not support the phenotypes we observed.

### Expression profile of unigenes involved in the anthraquinone biosynthesis pathway

Anthraquinones are bioactive compounds in *Rheum* plants. Their biosynthetic pathway is thought to involve the shikimate, MVA, MEP and polyketide pathways (*Leistner, 1971*). To identify the potential candidate unigenes of the anthraquinones biosynthetic pathway, we identified homologous genes by aligning the transcriptome sequence of *R. tanguticum* with all the known enzyme sequences associated with the above-mentioned pathways. We then screened these results by combining the sequence similarity and functional annotation to obtain the most conceivable candidate unigenes involved in the anthraquinones biosynthesis pathway. Using this process, we identified 79 unigenes associated with the anthraquinones biosynthesis pathway in *R. tanguticum* (Table S14). In the shikimate pathway, most *RtDAHPS*, *RtDHQS*, *RtMenE*, and *RtEPSP* genes were highly expressed in green leaves. Interestingly, in the MEP pathway, most *RtISPG* genes were highly expressed in purple samples. Other catalytic genes (*RtDXPS*, *RtDXR*, *RtISPD*, *RtISPF* and *RtISPH*) had varying expression levels in the green and purple plants. In the MVA pathway, *RtHMGR* genes showed tissue-specific expression regardless of color types in both green and purple petioles. The *R. tanguticum* rhizome is considered to be the highest quality of all medical rhubarb in medicinal material markets. We found that *RtHMGS*, *RtHMGR*, *RtMK*, *RtPMK* and *RtMPD* were all highly expressed in green rhizome samples. In the PKS pathway, *RtPKS* was highly expressed in green rhizomes and *RtPKC* expressions were higher in green leaves.

The regulatory network of the anthraquinone biosynthetic pathway in *R. tanguticum* showed differential expression patterns (Fig. 6A). The co-expression analysis of anthraquinone and TFs showed that *RtSMK, RtMK, RtDXPS, RtHMGR, RtHMGS, RtMenB, RtMPD,* and *RtSDH* gene family members have expression patterns that closely resemble multiple TFs, including bHLH, WRKY, MADS, and MYB TFs (Fig. 6B). We cannot rule out the possibility that anthraquinone is synthesized in the leaves and petioles of *R. tanguticum* and then transported below ground for long-term storage. In future studies, the transcriptional network involved in the regulation of anthraquinone biosynthesis and transportation should be investigated.

## DISCUSSION

Photosynthesis is a vital metabolic process which supports plant growth and development (*Evans, 2013*). Chlorophyll is a key component of photosynthesis which absorbs energy from sunlight to transfer it to other parts of the photosystem. Under strong light conditions, in order to protect against excess light absorption, plants alter their gene expressions to reduce the amount of light reaching the chloroplast and/or to counteract the production of reactive oxygen species (ROS). Thus, dynamic control of chlorophyll content depends on biosynthesis and degradation to ensure optimal photosynthesis and plant fitness (*Maunders & Brown, 1983*). In this study, we investigated the chlorophyll biosynthesis and

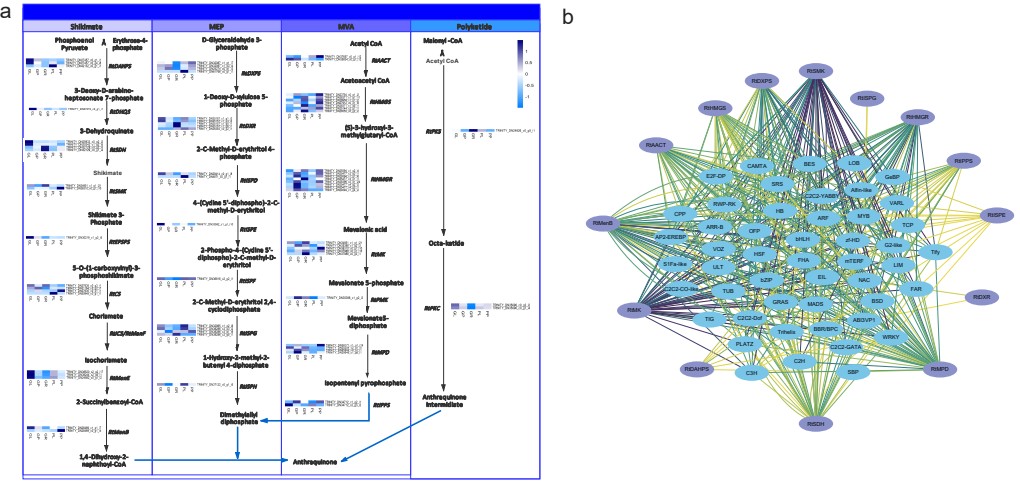

**Figure 6  Anthraquinone candidate pathway in _R. tanguticum_.** (A) The horizontal axis from left to right of each heatmap represents green leaves (GL), green petioles (GP), green rhizomes (GR), purple leaves (PL), and purple petioles (PP). (B) The co-expression network of anthraquinone genes and TFs.

degradation pathway genes in samples of _R. tanguticum_ collected from the Aba Tibetan and Qiang autonomous prefecture, which is located at a high elevation and has high UV radiation. In the chlorophyll biosynthesis pathway, most genes were up-regulated in the green leaves and petioles of _R. tanguticum_. Conversely, in the chlorophyll degradation pathway, _RtPPH_ was highly expressed in the purple samples, which suggests it breaks down pheophytin to pheophorbide A. The _RtPaO_ and _RtRCCR,_ which are involved in the red chlorophyll catabolite to primary fluorescent chlorophyll, were also highly expressed in the purple samples. The up-regulation of these three genes could intensify chlorophyll degradation in purple tissues. These results indicate that the regulation of chlorophyll content in purple tissues is controlled by both the biosynthesis and degradation pathways.

Anthocyanins are flavonoid pigments conferring red, blue, and purple colors to plant tissues. They can protect the leaf's photosynthetic system from damage (_Silva et al., 2016_) and help the plant be more resistant to stresses be regulating reactive oxygen signaling (_Hatier & Gould, 2008_). The anthocyanin biosynthesis pathway has been well described in many plants (_Winkel-Shirley, 2001_). We compared the expression levels of the genes involved in the anthocyanin pathway in different colored tissues. Anthocyanin content depends on the balance between biosynthesis and degradation (_Liu et al., 2018_). Most genes in the anthocyanin biosynthesis pathway were more highly expressed in green samples than purple samples. Published transcriptome studies about _Brassica juncea_ reveal that anthocyanin biosynthesis genes are more up-regulated in purple leaves than in green leaves (_Heng et al., 2020_). However, the stability of anthocyanins is dependent on the type of anthocyanin pigment, co-pigments, light, temperature, pH, metal ions, enzymes, oxygen, and antioxidants (_Turturica et al., 2015_). The role anthocyanins play in _R. tanguticum_ needs to be further explored.

We also evaluated the transcriptional changes of anthraquinone biosynthesis in the green and purple samples and found there was no prominent difference in the candidate genes of the anthraquinone pathway between green and purple *R. tanguticum*. We anticipated that the regulation of anthraquinone biosynthetic genes was not strongly associated with the plant colors. The amount of anthraquinone in different color tissues still needs to be further explored.

## CONCLUSIONS

This study analyzed the transcriptome profiles of purple and green samples in *R. tanguticum*. By comparing the FPKM values of green and purple *R. tanguticum*, we found that most chlorophyll biosynthesis genes were down-regulated in purple samples, and that the degradation pathway genes of chlorophyll (*RtPPH*, *RtPaO,* and *RtRCCR*) had higher expression levels in purple samples. In contrast, the anthocyanin biosynthesis enzymes (*e.g.*, ANS and UFGT) were more highly expressed in green samples than in the purple ones. Thus, these results indicate that the transcriptional regulation of chlorophyll metabolism plays more important roles in purple samples than the regulation of anthocyanins biosynthesis which contribute the color phenotypes in *R. tanguticum*. Although we also identified the anthraquinone biosynthesis pathway genes, we did not find any obvious relationship between the expression levels of these genes and plant colors.

### Funding

This work was supported by funding from the Major Science and Technology Projects of Yunnan Province (Digitalization, development and application of biotic resource, No. 860 202002AA100007, H.L.), the National Key R&D Program of China (No. 2019YFC1711000), the Shenzhen Municipal Government of China (grants JCYJ20170817145512476 and JCYJ20180507183534578), the Guangdong Provincial Key Laboratory of Genome Read and Write (grant 2017B030301011), and the NMPA Key Laboratory for the Rapid Testing Technology of Drugs. The funders had no role in study design, data collection and analysis, decision to publish, or preparation of the manuscript.

### Grant Disclosures

The following grant information was disclosed by the authors:
Major Science and Technology Projects of Yunnan Province: 60 202002AA100007.
National Key R&D Program of China:  2019YFC1711000.
Shenzhen Municipal Government of China:  JCYJ20170817145512476, JCYJ20180507183534578.
Guangdong Provincial Key Laboratory of Genome Read and Write: 2017B030301011.
NMPA Key Laboratory for the Rapid Testing Technology of Drugs.

## Competing Interests

The authors declare there are no competing interests. Haixia Chen, Tsan-Yu Chiu, Sunil Kumar Sahu, Haixi Sun, Jiawen Wen, Jianbo Sun, Qiyuan Li, Huan Liu are employed by BGI-Shenzhen. Yangfan Tang and Hong Jin are our partners.

## Author Contributions

- Haixia Chen conceived and designed the experiments, analyzed the data, prepared figures and/or tables, authored or reviewed drafts of the article, and approved the final draft.
- Tsan-Yu Chiu conceived and designed the experiments, prepared figures and/or tables, authored or reviewed drafts of the article, and approved the final draft.
- Sunil Kumar Sahu analyzed the data, prepared figures and/or tables, authored or reviewed drafts of the article, and approved the final draft.
- Haixi Sun analyzed the data, prepared figures and/or tables, authored or reviewed drafts of the article, and approved the final draft.
- Jiawen Wen performed the experiments, authored or reviewed drafts of the article, sample collection and processing, and approved the final draft.
- Jianbo Sun performed the experiments, authored or reviewed drafts of the article, upload raw data, and approved the final draft.
- Qiyuan Li performed the experiments, authored or reviewed drafts of the article, upload raw data, and approved the final draft.
- Yangfan Tang performed the experiments, authored or reviewed drafts of the article, sample collection and processing, and approved the final draft.
- Hong Jin conceived and designed the experiments, authored or reviewed drafts of the article, and approved the final draft.
- Huan Liu conceived and designed the experiments, authored or reviewed drafts of the article, and approved the final draft.

## DNA Deposition

The following information was supplied regarding the deposition of DNA sequences:

Transcriptome and unigene set of *R. tanguticum* using 19 samples are available at CNGBdb (China National GeneBank DataBase) (https://db.cngb.org/) with assembly accession CNA0022700 and CNA0022701, respectively.

## Data Availability

The raw data accessions of *R. tanguticum* transcriptome, CNS0095722–CNS0095726, CNS0095766, CNS0238596–CNS0238606, CNS0238637, CNS0238638 are available at CNP0000432.

## Supplemental Information

Supplemental information for this article can be found online at http://dx.doi.org/10.7717/peerj.14265#supplemental-information.

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
