# Peer review of "Transcriptomic analyses provide new insights into green and purple color pigmentation in Rheum tanguticum medicinal plants"

_PeerJ, doi:10.7717/peerj.14265_

## Round 0.1 · original submission · Major Revisions

Authors should follow observation of reviewers. qPCR validation is important when you have 3 or less replicates. Biochemical data is also very important to support RNAseq results.

·

Basic reporting

The paper very well organised and written. I think, the paper could be accepted after some improvements.

Experimental design

The experimental design is as it should be. However,

I think in this paper, the biggest missing part is qPCR. I think, the authors should do some qPCR analysis. For example, I am really wondering why anthocyanin related genes was higher in green tissues rather than purple tissues. If you do qPCR on some of anthocyanin related genes. We can check this.
Similarly, I think if the authors can do qPCR on some of the identified genes related to anthraquinones biosynthesis pathway, that would be much better for verification of RNA-seq.
In discussion part, I think the discussion should be improved. For example, the authors said that We anticipated that the regulation of anthraquinone biosynthetic genes was not strongly associated with the plant colors… What about previous researches. Have they given similar results??

Validity of the findings

The papers would be useful and beneficial for future researches.

Additional comments

I don't have any additional comments.

Reviewer 2 ·

Basic reporting

Rheum is very famous herbal genus all over the world, and R. tanguticum exists a variety of phenolic differences, such as the color changes in leaves. The authors generated and analyzed the RNA-seq data of R. tanguticum, including five green leaf samples, five green petiole samples, two green rhizome samples, four purple leaf samples, and three purple petiole samples, which showed that the purple pigmentation was mainly due to the effects of chlorophyll degradation. However, the manuscript needs to be polished and added more expermental data.

Previous studies have revealed the adaptive strategies of R. austral to face the harsh enviromental conditions. Here, the authors provided the RNA-seq data of R. tanguticum to understand the gene regulatory networks under the harsh enviromental conditions. But in may opinion, these data is difficult to fulfill this goal. The enviromental data of the sample collection should firstly be provided.

The authors performed the expression patterns of the genes involved in chlorophyll, anthocyanin and anthraquinone biosynthesis pathways. But this manuscript's data is too limited, the content of chlorophyll, anthocyanin and anthraquinone in those samples should be measured. In addition, the expression levels of the DEGs, such as RtPPH, RtPaO..., should be tested by qPCR or other experiments, not only the RNA-seq data.

Also, the figure quality and English of this manuscript should be improved.

Experimental design

Why the biological duplications between the five tissues are so different? And even the green rhizome tissue has only two duplications.

Validity of the findings

no comment.

Additional comments

There are some minor errors in this manuscript, such as:
1. P47-48: “Dahuang Rhei Radix et” should not be italic.
2. P49: “Rheum” should be italic.
....
English language should be improved.

Reviewer 3 ·

Basic reporting

(1) The format of the specie name (Rheum tanguticum Maxim. ex Balf.) and English grammar might be checked in the text to ensure other audiences can clearly understand.
(2) The background about plant color research was not enough, which might be improved.
(3) Line 409-412: There is more than one assembly version under the project accession of CNP0001659, could you please explain it? And the deposited data couldn’t be accessed with the assembly accessions and raw data accessions, wouldn’t be downloaded. The expression data of this paper should be uploaded.

Experimental design

(1) This article focused on chlorophyll, anthocyanin, and anthraquinone pathway to explain plant color changing, it’s an impressive perspective, but some details might be improved in the background and results.
(2) The experimental design is generally good, but there are two green rhizomes samples without a purple samples comparison, is it necessary for this article in this experiment design?

Validity of the findings

(1) RNA-seq data might be verified with qRT-PCR.
(2) Line100-102, the description might be incorrect, there is significantly different terminology used between a de-novo assembly and assembly with a genome.
(3) Line106-110, sentences in this paragraph might be checked and reconstructed.
(4) Line159-160, the description of the PCA results makes comprehension difficult, and what are the two data set means?
(5) Line 228-231, The paragraph about the anthocyanin pathway might be described in more detail.
(6) Are there any relationships between anthraquinones and plant colors (green or purple)?
(7) Line 255-262, this paragraph seems to describe the statistical result, do they have any roles in explaining the plant color changing in this research. is it necessary for this article?
(8) The part of the conclusion was not enough to summarize and clarify the biological questions.

Additional comments

no comment

---

## Round 0.2 · Minor Revisions

Authors still raises important issues.

Reviewer 2 ·

Basic reporting

Due to the sample collection limitation, the authors could not carry out qPCR experiments. Recent manuscript could draft the transcriptional network in the regulation and biosynthesis of medicinally active compounds in R. tanguticum.

Experimental design

No comment.

Validity of the findings

No comment.

Additional comments

1. Language of this draft should be improved.
2. The raw data of the transcripton should be submitted before this paper could be accepted.

---

## Round 0.3 · Minor Revisions

Authors addressed most reviewers issues and the article can be editorially accepted. Please have the manuscript checked for readability before final acceptance.

·

Basic reporting

It is acceptable.

Experimental design

It is acceptable.

Validity of the findings

It is acceptable.

Additional comments

No comment.

Reviewer 2 ·

Basic reporting

No comment.

Experimental design

No comment.

Validity of the findings

No comment.

Additional comments

No comment.

---

## Round 0.4 · accepted · Accept

In this version, the author improved the readability of the manuscript.